# *Pneumocystis jirovecii* pneumonia mortality risk associated with preceding long-term steroid use for the underlying disease: A multicenter, retrospective cohort study

**Kohei Miyake**[1]\*, **Satoru Senoo**[2☯¤], **Ritsuya Shiiba**[3☯], **Junko Itano**[4☯], **Goro Kimura**[4‡], **Tatsuyuki Kawahara**[5☯], **Tomoki Tamura**[6☯], **Kenichiro Kudo**[7☯], **Tetsuji Kawamura**[1‡], **Yasuharu Nakahara**[1‡], **Hisao Higo**[2‡], **Daisuke Himeji**[3‡], **Nagio Takigawa**[5‡], **Nobuaki Miyahara**[2‡], **Okayama Respiratory Disease Study Group (ORDSG)**[¶]

**1** Department of Respiratory Medicine, National Hospital Organization Himeji Medical Center, Himeji, Japan, **2** Department of Hematology, Oncology and Respiratory Medicine, Okayama University Graduate School of Medicine, Dentistry, and Pharmaceutical Sciences, Okayama, Japan, **3** Department of Internal Medicine, Miyazaki Prefectural Miyazaki Hospital, Miyazaki, Japan, **4** Department of Allergy and Respiratory Medicine, National Hospital Organization Minami-Okayama Medical Center, Okayama, Japan, **5** Department of General Internal Medicine 4, Kawasaki Medical School, Okayama, Japan, **6** Department of Respiratory Medicine, National Hospital Organization Iwakuni Clinical Center, Iwakuni, Japan, **7** Department of Respiratory Medicine, National Hospital Organization Okayama Medical Center, Okayama, Japan

☯ These authors contributed equally to this work.
¤ Current address: Department of Respiratory Medicine, National Hospital Organization Fukuyama Medical Center, Fukuyama, Japan
‡ GK, TK, YN, HH, DH, NT, and NM also contributed equally to this work.
¶ Membership of the Okayama Respiratory Disease Study Group is provided in the Acknowledgments.
\* miyakou1723@yahoo.co.jp

## Abstract

### Objective

Long-term steroid use increases the risk of developing Pneumocystis pneumonia (PcP), but there are limited reports on the relation of long-term steroid and PcP mortality.

### Methods

Retrospective multicenter study to identify risk factors for PcP mortality, including average steroid dose before the first visit for PcP in non-human immunodeficiency virus (HIV)-PcP patients. We generated receiver operating characteristic (ROC) curves for 90-day all-cause mortality and the mean daily steroid dose per unit body weight in the preceding 10 to 90 days in 10-day increments. Patients were dichotomized by 90-day mortality and propensity score-based stabilized inverse probability of treatment weighting (IPTW) adjusted covariates of age, sex, and underlying disease. Multivariate analysis with logistic regression assessed whether long-term corticosteroid use affected outcome.

**Data Availability Statement:** All relevant data are within the paper and its Supporting information files.

**Funding:** The author(s) received no specific funding for this work.

**Competing interests:** K.M. have received personal fees from Nippon Shinyaku Co td and AstraZeneca. N.T. have received grants and personal fees from Eli Lilly Japan, AstraZeneca, Daiichi-Sankyo Pharmaceutical, Chugai Pharmaceutical, Taiho Pharmaceutical, Pfizer Inc. Japan, Boehringer-Ingelheim Japan, and Ono Pharmaceutical; grants from Kyowa Hakko Kirin, Nippon Kayaku Co. Ltd., and Takeda Pharmaceutical Co. Ltd.; and personal fees from MSD and Bristol-Myers Squibb Company Japan outside the submitted work. None declared (S.S., R. S., J. I., G. K., T. K., T. T., K. K., T. K., Y. N., H. H., D. H., N. M.) This does not alter our adherence to PLOS ONE policies on sharing data and materials.

## Results

Of 133 patients with non-HIV-PcP, 37 died within 90 days of initial diagnosis. The area under the ROC curve for 1–40 days was highest, and the optimal cutoff point of median adjunctive corticosteroid dosage was 0.34 mg/kg/day. Past steroid dose, underlying interstitial lung disease and emphysema, lower serum albumin and lower lymphocyte count, higher lactate dehydrogenase, use of therapeutic pentamidine and therapeutic high-dose steroids were all significantly associated with mortality. Underlying autoimmune disease, past immunosuppressant use, and a longer time from onset to start of treatment, were associated lower mortality. Logistic regression analysis after adjusting for age, sex, and underlying disease with IPTW revealed that steroid dose 1–40 days before the first visit for PcP (per 0.1 mg/kg/day increment, odds ratio 1.36 [95% confidence interval = 1.16–1.66], $P<0.001$), low lymphocyte counts, and high lactate dehydrogenase revel were independent mortality risk factor, while respiratory failure, early steroid, and sulfamethoxazole/trimethoprim for PcP treatment did not.

## Conclusion

A steroid dose before PcP onset was strongly associated with 90-day mortality in non-HIV-PcP patients, emphasizing the importance of appropriate prophylaxis especially in this population.

## Introduction

Pneumocystis pneumonia (PcP), caused by the yeast-like fungus *Pneumocystis jirovecii*, occurs primarily in immunocompromised hosts [1]. Historically, PcP was initially found in human immunodeficiency virus (HIV) patients but has increased among non-HIV subjects in parallel with the increased use of immunosuppressive drugs including corticosteroids [2–5]. PcP can lead to respiratory failure, a potentially life-threatening condition; the reported mortality rate for non-HIV-PcP is higher than for HIV-associated PcP (HIV-PcP) [5–7]. In non-HIV-PcP, higher doses of corticosteroids are reported to be the most important risk factor [3, 8], and in patients with systemic autoimmune diseases, higher doses of corticosteroids are reported to increase the risk of developing PcP [9]. In guidelines for treating hematologic and solid tumors [10, 11], primary prevention of PcP is recommended for patients with prolonged steroid use (>20 mg/day equivalent of prednisone for 4 weeks or ≥1 month). Although the efficacy of sulfamethoxazole/trimethoprim (TMP-SMX) prophylaxis for the prevention of PcP is well established [12–15], long-term administration of TMP-SMX can cause a variety of side effects, including thrombocytopenia, toxemia, liver dysfunction, and electrolyte abnormalities [16]. The therapeutic effect of corticosteroids on PcP is complex. These drugs are well-established as therapy for HIV-PcP patients, blunting the inflammatory response induced by anti-pneumocystis treatment, preventing respiratory decompensation failure, and decreasing mortality [17, 18]. However, in non-HIV patients, early addition of corticosteroids to anti-pneumocystis therapy was not associated with improved survival outcomes [19–26]. Furthermore, it has even been reported that high steroid doses at diagnosis [27] or long-term pre-onset steroids [28, 29] were predictors of poor prognosis.

Corticosteroid use for underlying disease may affect mortality risk in PcP [27–30], but studies of dose and duration that affect mortality are lacking. Details on the duration and weight-

corrected intensity of steroid loading are also needed, given considerations of average body weight due to racial differences, as well as its mechanism of action in causing late immunosuppression. We therefore studied the impact of long-term corticosteroid treatment for underlying disease on mortality of PcP patients. This retrospective observational study aimed to identify risk factors for poor prognosis in non-HIV-PcP, with a particular focus on the weight-corrected intensity and duration of corticosteroid therapy received prior to the onset of PcP.

## Materials and method

### Study design

This retrospective, multicenter observational study was conducted by the Okayama Respiratory Disease Study Group. All subjects in this retrospective study were Japanese. Adult patients (age 18 years and older) diagnosed with non-HIV-PcP between August 1, 2010, and August 31, 2022, were included in the retrospective review of medical records at seven Japanese sites. The data collection period was from February, 10, 2022 to March, 31, 2023. The primary endpoint of this study was 90-day all-cause mortality, based on previous studies [21, 26, 31, 32].

The study was approved by the Institutional Ethics Committee of National Hospital Organization Himeji Medical Center (No. 2020–28) and all other participating hospitals. The data were analyzed anonymously. Informed consent was obtained on an opt-out basis, and a description of the study was displayed on the websites of Himeji Medical Center and each center. The requirement for written informed consent was waived because of the retrospective nature of the study. The study was conducted in accordance with the Declaration of Helsinki and all relevant Japanese laws and regulations.

### Case definition

Eligible patients were defined according to their adherence to all of the following selection criteria: (1) symptoms consistent with PcP (i.e., dyspnea, fever, and/or cough); (2) evidence of new onset or progressive bilateral pneumonia on chest X-ray or computed tomography; (3) Detection of *P. jirovecii* cysts by Grocott staining or positive *P. jirovecii* DNA-specific qualitative PCR (*P.jirovecii* DNA 6B614-0000-061-851, SRL, Inc., Shinjuku, Japan [33]) in respiratory specimens; (4) Exclusion of patients who were diagnosed with other types of infectious or autoimmune pneumonia by sputum culture, nucleic acid test, and antibody measurement.; (5) Exclusion of HIV-positive patients, and patients with insufficient data (weight, drug use data for underlying disease, etc.). The 90-day all-cause mortality was calculated from the first day of medical care for PcP.

### Clinical data

Data extracted from electronic medical records included patient demographic characteristics, underlying disease, previous immunosuppressive regimens for the underlying disease (including corticosteroids, immunosuppressive drugs, anticancer drugs, and biologics), clinical findings such as respiratory failure with a resting peripheral oxygen saturation ($SpO_2$) < 90% on admission, laboratory findings on admission, anti-pneumocystis treatment regimen, and course of treatment. In addition, details of all corticosteroids use during hospitalization were collected, including drug name, daily dose, and duration of administration. Steroid doses were expressed in prednisolone (PSL) equivalents as follows [34]: hydrocortisone, 0.25 mg PSL per mg; methylprednisolone, 1.25 mg PSL per mg; betamethasone and dexamethasone, 7.5 mg PSL per mg. For TMP-SMX, doses were calculated using the trimethoprim equivalent. For immunosuppressants, drugs administered during 90 days prior to the first PcP visit, and for

administration of biologics and anticancer drugs, use or no use over the 90-day period was recorded. For PcP treatment, the average daily dose of PSL per unit body weight was calculated from the day of the first visit to day 5. For TMP-SMX, the trimethoprim equivalent daily dose per unit body weight was tabulated, and the use or lack use of atovaquone and pentamidine during the period was tabulated.

## Statistical analysis

Continuous variables were summarized by median interquartile range (IQR). Descriptive analysis was used to examine patient characteristics. A receiver operating characteristic (ROC) curve was constructed with 90 days PcP mortality as a conditional variable and the mean daily steroid dose per unit body weight in the past 10 days to 90 days in increments of 10 days prior to the date of the first visit for PcP. The time with the highest area under the ROC curve (AUC) was adopted as the period which most affected mortality, and the mean steroid consumption during this period was analyzed. Subsequently, we categorized the average steroid dose during this period and investigated the association with 90-day mortality using logistic regression analysis. Next, to estimate the risk of PcP mortality, we examined the average steroid dose during the period of maximum AUC, as well as the use of other drugs within 30 days prior to the first visit for PcP after initial diagnosis. For the PcP treatment period, the mean daily dose of corticosteroid per unit body weight and TMP-SMX were calculated, divided into 5 day segments; for the other drugs, only use or lack of use during the period in question was considered. The outcome of interest was 90-day all-cause mortality, and the χ-square test was used for testing categorical variables and the Mann-Whitney U test for continuous variables to determine significant risk factors related to PcP mortality.

To account for the possibility of selection bias in this observational study (since the intensity of corticosteroid treatment for the underlying disease was at the discretion of the attending physician), we used an analysis method including stabilized treatment-weighted inverse probability of treatment weighting (IPTW) based on propensity scores. The propensity score was defined as baseline covariates (age [21, 35–38], sex, underlying disease) previously described as heightening the risk of death (hematologic disease [21, 22, 31], solid organ malignancy [39, 40], and interstitial lung disease [ILD] [38, 41]). Using an IPTW method with truncated stabilization weights, we reconstructed a pseudo-population in which patients in both groups (90-day deceased vs. 90-day survival) had similar characteristics balanced for age, sex, and underlying disease covariates, so that outcomes could be meaningfully compared.

Adjusted data were then used to evaluate the risk of death among the 90-day survival or mortality groups using logistic regression. Explanatory variables were included in the propensity score model, including steroid burden for the underlying disease during the period of maximum AUC, days from onset to start of treatment [36, 40, 42], worsening respiratory status [30, 40, 43, 44], lactate dehydrogenase (LDH) [29, 40], lymphocyte count [35, 42, 45], and early treatment with steroids, and TMP-SMX, which had previously been implicated in increasing the risk of death. Missing values were not complemented. EZR [46] was used for statistical analysis, and *P* <0.05 was considered significant.

## Result

### Patient characteristics

During the 11-year study period, 142 patients were diagnosed with PcP at seven centers, and 133 patients met the criteria and were selected for the study (Fig 1). The median (IQR) age of all patients was 71.5 (64.4–78.5) years, and 49 (36.8%) were female. Total mortality at 90 days after the first visit was 37 (27.8%) patients (Table 1). With the exception of 3 patients, the

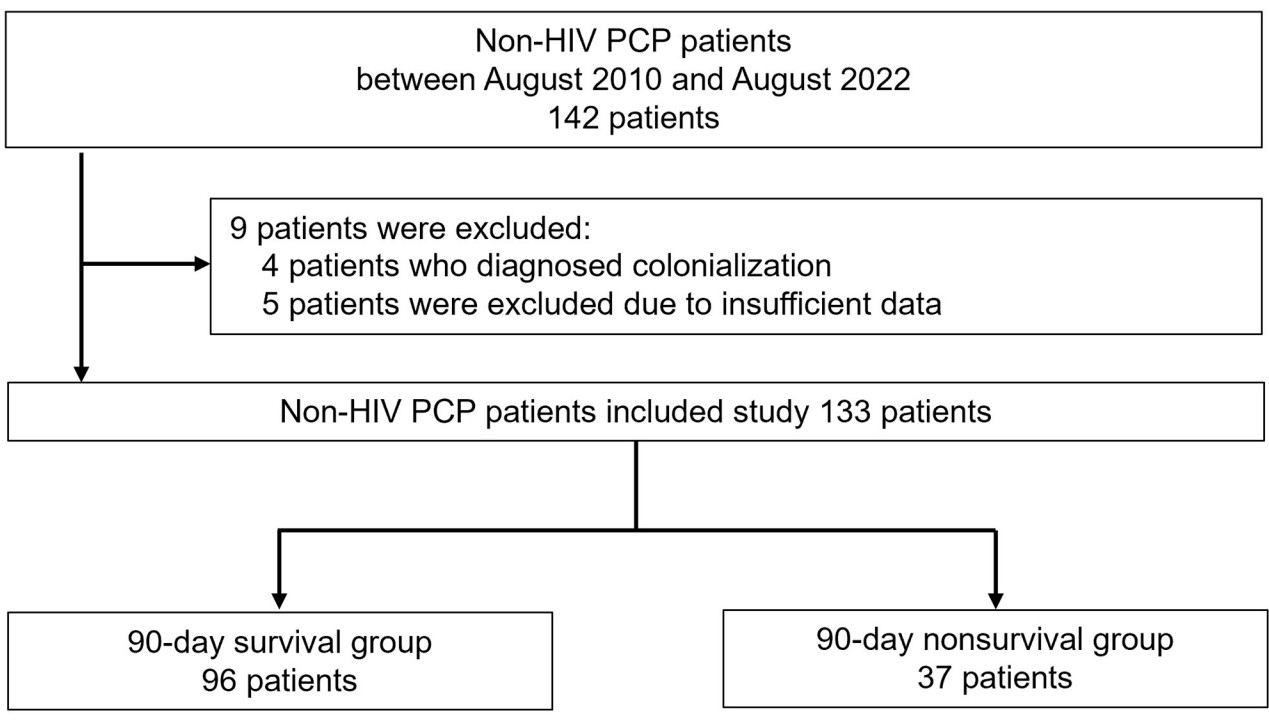

**Fig 1. Patient selection flowchart.**

majority (130 patients) had received immunosuppressive drugs or had immunosuppressive disease at the time of diagnosis of PcP. The most common underlying disease type was autoimmune (N = 61), followed by solid organ malignancy (N = 43) and blood disease (N = 24). Underlying pulmonary disease was also present in 94 patients, with ILD (N = 60) and emphysema (N = 49) accounting for the majority of cases. There were 126 patients (94.7%) with a history of immunosuppressive or anticancer drug use in the 90 days prior to the first visit, the most common of which was corticosteroid in 101 patients (75.9%), with a median (IQR) dose of 0.24 (0.10–0.43) mg/kg/day. The next most common group was immunosuppressant users (N = 58), followed by anticancer drug users (N = 44). Six patients received PcP prophylaxis in the month of the first visit; one with pentamidine and the remaining five with TMX-SMP, but all at reduced doses than specified in the guidelines [47]. The median (IQR) time from onset of PcP symptoms to the first visit was 4 (0–7) days; 75 patients (56.4%) had respiratory failure with $SpO_2 < 90\%$ (room air) at the first visit. Treatment consisted of antimicrobials in 130 patients (TMX-SMP in 124, atovaquone in 23, pentamidine in 18, with concomitant use), and 2 of 3 patients who did not use antibiotics throughout the whole course died. 24 patients did not use antibiotics on days 1–5. Corticosteroids were started or increased in 95 patients over the whole course up to 90 days after the initial visit. The median (IQR) number of days from initial visit to antimicrobial initiation was 1(0–3) day and the median (IQR) number of days to steroid initiation or dose escalation was 1(0–3) day.

### ROC curves for corticosteroid dosage

The AUCs of each ROC curve constructed with the mean daily corticosteroid use per unit body weight and 90-day mortality in the past 10 days to 90 days in increments of 10 days are shown in Table 2. The AUC was the highest for the past 1–40 days for 90-day mortality

**Table 1. Baseline patient and demographic characteristics.**

| | All patients (N = 133) | N/A | Non-survivors (N = 37) | Survivors (N = 96) | *P* value [a] | OR(CI) |
|---|---|---|---|---|---|---|
| age | 71.5 (64.4–78.5) | 0 | 71.4 (67.2–79.4) | 71.6 (61.6–77.7) | 0.255 | |
| male sex | 84 | 0 | 25 | 59 | 0.650 | 1.30 (0.55–3.21) |
| smoking Pack Years | 16.8 (0–42) | 8 | 26.2 (0–48) | 15 (0–40) | 0.444 | |
| body weight (kg) | 54.0 (47.0–60.0) | 0 | 54.7 (48.8–58.8) | 53.5 (46.6–60.1) | 0.798 | |
| height (cm) | 159.9 (151.6–165.0) | 6 | 159.5 (151.6–164.8) | 160.0 (151.8–165.0) | 0.857 | |
| basal disease | | | | | | |
| autoimmune disease | 61 | 0 | 11 | 50 | 0.034 | 0.39 (0.16–0.93) |
| solid organ tumor | 43 | 0 | 15 | 28 | 0.294 | 1.65 (0.68–3.90) |
| hematologic disease | 24 | 0 | 7 | 17 | 0.679 | 1.38 (0.46–3.86) |
| solid organ transplantation | 4 | 0 | 0 | 4 | 0.506 | 0.00 (0.00–3.94) |
| interstitial lung disease | 60 | 0 | 23 | 37 | 0.024 | 2.60 (1.12–6.21) |
| Asthma | 2 | 0 | 0 | 2 | 0.929 | 0.00 (0.00–13.9) |
| basal lung CT findings | | | | | | |
| bronchiectasis | 17 | 1 | 6 | 11 | 0.560 | 1.54 (0.43–5.03) |
| emphysema | 49 | 1 | 20 | 29 | 0.013 | 2.86 (1.22–6.86) |
| honeycomb lung | 14 | 1 | 4 | 10 | 1.000 | 1.07 (0.23–4.31) |
| traction bronchiectasis | 44 | 1 | 17 | 27 | 0.062 | 2.27 (0.96–5.41) |
| chronic immunosuppressive regimen | | | | | | |
| basal corticosteroid dose | | | | | | |
| 1–30 day [b] | 0.126 (0.035–0.377) | 0 | 0.412 (0.253–0.618) | 0.096 (0.000–0.193) | <0.001 | |
| 1–60 day [b] | 0.123 (0.050–0.380) | 0 | 0.421 (0.281–0.629) | 0.097 (0.095–0.218) | <0.001 | |
| 1–40 day [b] | 0.119 (0.039–0.393) | 0 | 0.425 (0.336–0.695) | 0.093 (0.000–0.183) | <0.001 | |
| basal immunosuppressive regimen | | | | | | |
| immunosuppressant | 58 | 0 | 10 | 48 | 0.028 | 0.37 (0.14–0.90) |
| MTX | 39 | 0 | 3 | 36 | 0.002 | 0.15 (0.03–0.53) |
| CNI | 19 | 0 | 6 | 13 | 0.906 | 1.23 (0.35–3.87) |
| orther immunosuppuressant | 9 | 0 | 4 | 5 | 0.443 | 2.19 (0.41–10.9) |
| biologic agent | 7 | 0 | 0 | 7 | 0.210 | 0.00 (0.00–1.77) |
| anticancer drug | 44 | 0 | 14 | 30 | 0.604 | 1.34 (0.55–3.16) |
| PcP prophylaxis | 6 | 0 | 4 | 2 | 0.088 | 5.61 (0.76–64.8) |
| onset to first touch (days) | 4 (0–7) | 0 | 2 (1–5) | 5 (0–9) | 0.062 | |
| Hypoxia | 75 | 0 | 29 | 46 | 0.003 | 3.90 (1.54–10.9) |
| laboratory data | | | | | | |
| (1→3) β-D-glucan [c] (pg/mL) | 31.3 (10.9–98.8) | 1 | 41.0 (15.7–141.8) | 27.6 (10.5–67.3) | 0.258 | |
| serum albumin (g/dL) | 2.9 (2.4–3.3) | 38 | 2.6 (2.3–3.2) | 3.0 (2.6–3.5) | 0.027 | |
| CRP (mg/dL) | 6.97 (4.03–11.46) | 0 | 6.59 (4.51–13.4) | 6.99 (3.73–10.8) | 0.825 | |
| KL-6 (U/mL) | 603 (370–970) | 17 | 827 (556–1645) | 534 (362–793) | 0.001 | |
| LDH (IU/L) | 330 (273–450) | 0 | 363 (310–538) | 326 (268–420) | 0.015 | |
| lymphocyte count (mm$^3$) | 650 (377–1,070) | 0 | 460 (317–680) | 720 (411–1185) | 0.004 | |
| SP-D (ng/mL) | 147 (93–239) | 57 | 236 (146–312) | 137 (78–223) | 0.027 | |
| white blood cell count (mm$^3$) | 8700 (6200–11000) | 0 | 7960 (5600–9400) | 8750 (6625–11723) | 0.060 | |
| PcP therapy time days | | | | | | |
| onset to anti PcP drug initiation | 6 (2–12) | 3 | 3 (2–8) | 7 (4–14) | 0.005 | |
| onset to steroid dose-up or initiation | 6 (2–11) | 38 | 2 (2–7) | 7 (3–11) | 0.012 | |
| first visit to anti-PcP drug initiation | 1 (0–3) | 3 | 1 (0–2) | 1 (0–4) | 0.184 | |
| first visit to steroid dose-up or initiation | 1 (0–3) | 38 | 1 (0–2) | 1 (0–3) | 0.604 | |

*(Continued)*

**Table 1.** (Continued)

| | All patients (N = 133) | N/A | Non-survivors (N = 37) | Survivors (N = 96) | *P* value [a] | OR(CI) |
|---|---|---|---|---|---|---|
| PcP therapy | | | | | | |
| corticosteroid dose for 1-5day** | 0.804 (0.122–8.001) | 0 | 2.441 (0.535–11.67) | 0.603 (0.078–6.653) | 0.015 | |
| TMP-SMX use for day 1–5 | 100 | 0 | 26 | 74 | 0.554 | 0.70 (0.28–1.84) |
| TMP-SMX dose for 1-5day** | 5.858 (0.292–10.505) | 0 | 4.683 (0.000–10.105) | 5.998 (0.854–10.936) | 0.549 | |
| atovaquone use for day 1–5 | 6 | 0 | 2 | 4 | 1.000 | 1.31 (0.11–9.62) |
| pentamidine use for day 1–5 | 8 | 0 | 7 | 1 | <0.001 | 21.6 (2.61–103) |
| first visit to last follow days | 124 (24–954) | 0 | 17 (8–32) | 422 (96–1357) | | |

CI: confidence interval; NA: not available; OR: odds ratio,

Continuous variables were expressed as median (IQR), and nominal variables were expressed as number of cases.

[a]Mann-Whitney U or χ-square test

[b]mg/kg/day

[c]122 patients underwent test WAKO (FUJIFILM Wako Pure Chemical Corp.) or 11 patients underwent Fungitec G Test MK II (NISSUI BG))

(0.8305, 95% confidence interval [CI] 0.7404–0.9206), and the distance-to-the-corner method indicated a cutoff value of mean corticosteroid dose of 0.34 mg/kg/day (Fig 2). Indeed, the mortality by dose category of cumulative steroid dose in the past 1–40 days, mortality was significantly higher above 0.3 mg/kg/day (Table 3, Fig 3).

## Comparison of survivors vs. non-survivors

A comparison of survivors (N = 96) and non-survivors (N = 37) is shown in Table 1. There were no significant differences in age and gender between the two groups. In terms of underlying disease, solid organ malignancies were significantly more common in the deceased patients, while autoimmune diseases were less common. Of the underlying pulmonary diseases, ILD and emphysema were significantly more common in the deceased patients with CT findings. Regarding medications for underlying disease, baseline corticosteroid daily doses per unit body weight for 30 and 60 days before the PcP first visit were significantly higher in the deceased patient group than in the survivors. Among immunosuppressive drugs, methotrexate (MTX) was used for significantly fewer patients in the deceased group, but no difference was seen for anticancer drugs. Respiratory failure was associated with a worse prognosis. Laboratory values of LDH, Krebs von den Lungen-6 (KL-6), and pulmonary surfactant protein-D

**Table 2.** AUC of ROC curve of increments of 10 days median corticosteroid dose for underlying disease in predicting 90-day all-cause mortality in patients with Pneumocystis pneumonia by period before the first visit.

| | AUC |
|---|---|
| 1–10 day | 0.800 |
| 1–20 day | 0.793 |
| 1–30 day | 0.815 |
| 1–40 day | 0.824 |
| 1–50 day | 0.818 |
| 1–60 day | 0.795 |
| 1–70 day | 0.778 |
| 1–80 day | 0.773 |
| 1–90 day | 0.769 |

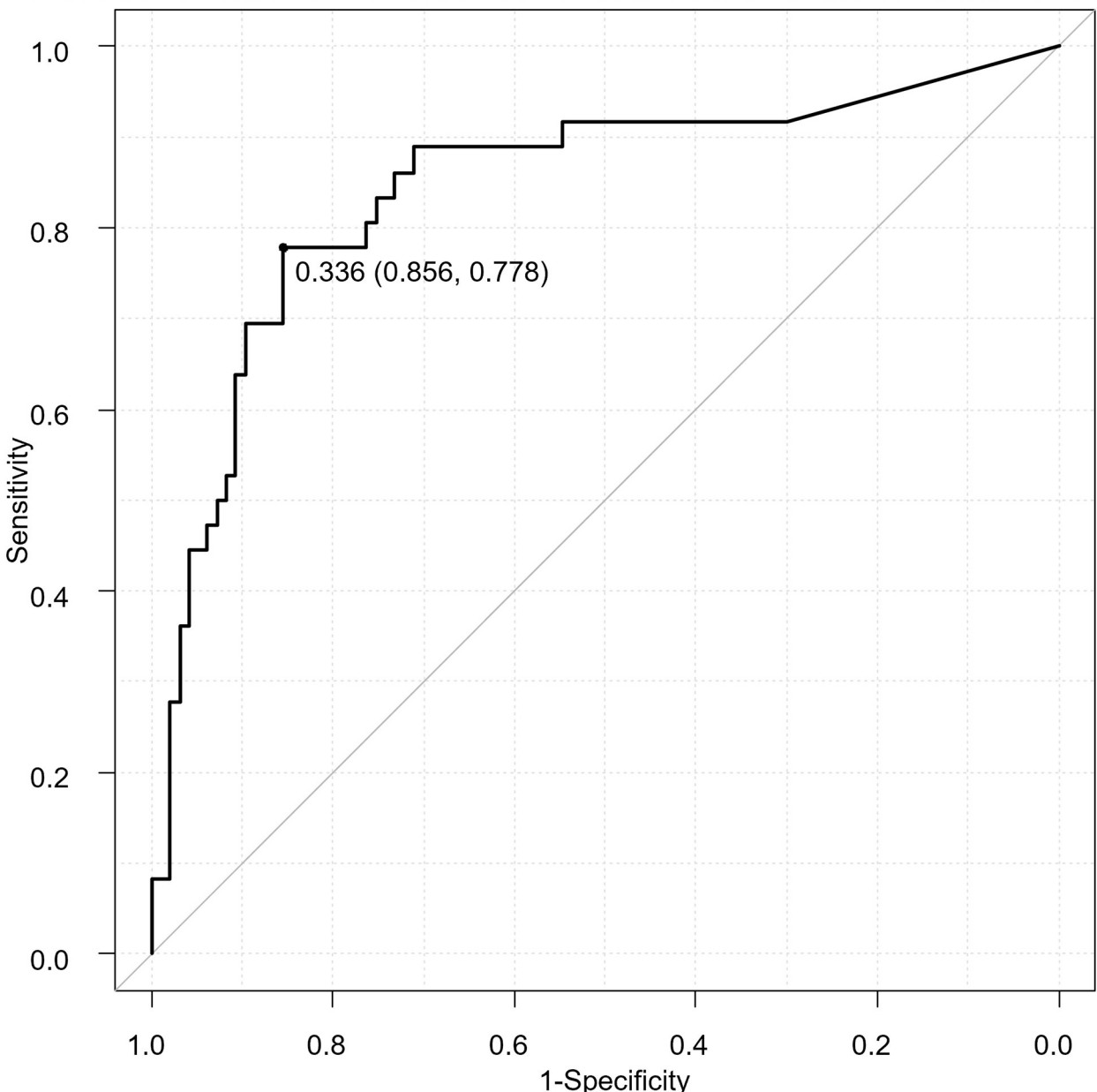

**Fig 2. ROC/AUC analysis.** ROC curve analysis showing the performance of past 40 days corticosteroid dose before PcP first visit in predicting all-cause 90-day mortality of non-HIV-PcP patients.

(SP-D) were significantly higher in the deceased patients, and serum albumin (Alb) and lymphocyte counts were significantly lower. The time from disease onset to the first medical examination was significantly shorter for the patients who died, and the days from disease onset to the start of treatment were also significantly fewer, and more deceased patients had progressed rapidly. In terms of treatment, the initial steroid dose of day 1–5 was significantly higher in the deceased group, but there was no significant difference in the fixed dose of TMX-SMP between survivors and non-survivors. Characteristics before and after the weighting process are shown in Table 4. In a multivariate analysis adjusted for propensity score reciprocal weighting, initial

**Table 3. 90-day mortality by dose category of cumulative steroid dose.**

| 1-40day corticosteroid dose | N | OR (95%CI) | *P* value |
|---|---|---|---|
| <0.1 [a] | 57 | Reference | Reference |
| 0.1–0.2 [a] | 24 | 1.49 (0.33–6.78) | 0.609 |
| 0.2–0.3 [a] | 6 | 2.08 (0.20–21.5) | 0.539 |
| 0.3–0.4 [a] | 14 | 5.78 (1.39–24.1) | 0.016 |
| 0.4–0.5 [a] | 14 | 18.7 (4.49–78) | <0.001 |
| ≥0.5 [a] | 18 | 36.4 (8.61–54) | <0.001 |

[a] mg/kg/day

CI: confidence interval; OR: odds ratio

therapeutic steroid dose and the fixed-dose combination were not significant markers, but mean steroid dose prior to PcP first day (per 0.1 mg/kg/day increment, odds ratio 1.36 [95% CI = 1.16–1.66], *P*<0.001), cumulative steroid dose, LDH, and lymphocyte count were significant predictors of mortality (Table 5).

## Discussion

In this retrospective study, we found that a corticosteroid dose before PcP onset had significant negative impact on 90-day all-cause mortality. Long-term corticosteroid loading for underlying disease increased the mortality of PcP in a dose-dependent manner. Furthermore, in

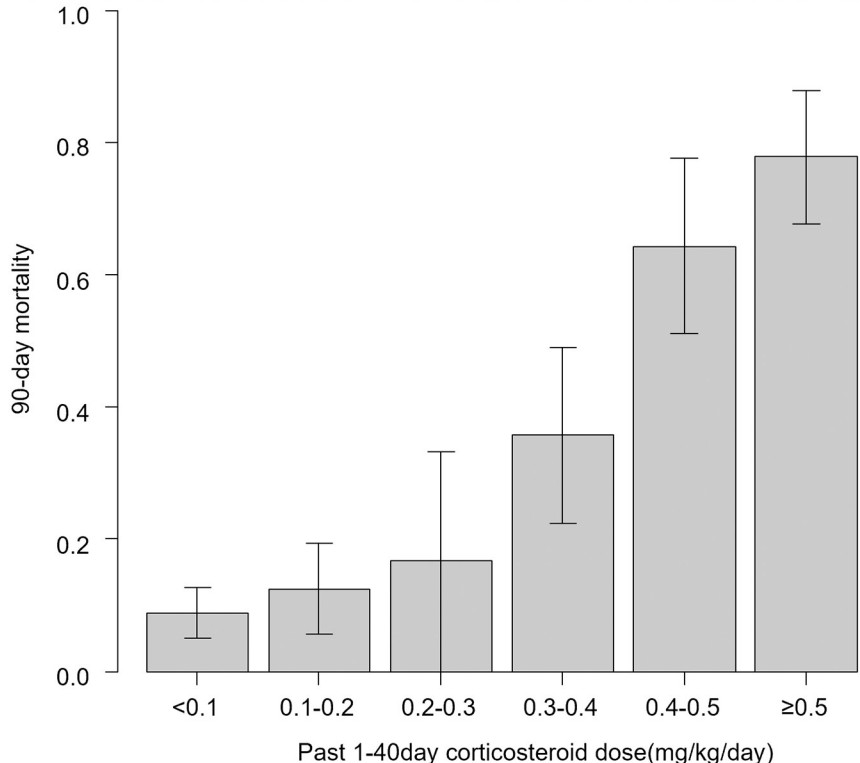

**Fig 3. 90-day mortality by category of mean steroid load 1–40 days before first visit.**

**Table 4. Propensity score-based stabilized treatment-weighted inverse probability inverse probability (IPTW) to adjust for the covariates age, sex and underlying disease.**

| mortality | Unweighted | | | IPTW-weighted | | |
|---|---|---|---|---|---|---|
| Factor | Nonsurvivors | Survivors | SMD | Nonsurvivors | Survivors | SMD |
| | 37 | 96 | | 34.9 | 96.8 | |
| Age (range) | 71.4 (46.8–90.0) | 71.6 (18.2–90.2) | 0.343 | 70.9 (46.8–90.0) | 72.2 (18.2–90.2) | 0.110 |
| Sex | | | | | | |
| M (%) | 25 (67.6) | 59 (61.5) | 0.128 | 22.5 (64.5) | 61.8 (63.8) | 0.014 |
| F (%) | 12 (30.6) | 37 (38.5) | | 12.4 (35.5) | 35.0 (36.2) | |
| Based disease | | | | | | |
| blood disease (%) | 8 (21.6) | 16 (16.7) | 0.126 | 7.9 (22.7) | 18.7 (19.3) | 0.085 |
| ILD (%) | 23 (62.2) | 37 (38.5) | 0.486 | 17.0 (48.8) | 44.3 (45.8) | 0.061 |
| Solid organ tumor (%) | 15 (40.5) | 28 (29.2) | 0.240 | 11.8 (33.9) | 31.6 (32.6) | 0.028 |

ILD: interstitial lung disease; SMD: standardized mean difference

multivariate analysis, long-term steroid loading before PcP onset significantly affected 90-day mortality, while respiratory failure, early steroid and TMP-SMX for PcP treatment did not. Pre-onset long-term steroid loading had the greatest impact. Although there is some doubt about the validity of calculating statistically daily, the AUC was maximized (0.8314) for the past 7–39 days. This suggests that corticosteroid loading in the more distant past than just before the onset of PcP has a greater impact on the mortality rate of PcP. Prior studies had shown that corticosteroids for underlying disease have an impact on the mortality of PcP [27–30]. Here, we report that baseline steroid loading was more strongly associated with the mortality of PcP than hypoxemia at the initiation of treatment or the corticosteroid and antibiotic treatment given for PcP. This suggests that proper risk assessment and appropriate prophylaxis in high-risk patients is more important for PcP than early detection and treatment. However, in the present study, patients who developed PcP despite prophylaxis was not good prognosis, so further investigation is needed on the intensity and duration of such prophylaxis.

**Table 5. Multivariate logistic regression analysis.**

| | Unadjusted | | Propensity weighted | |
|---|---|---|---|---|
| | OR (95% CI) | *P* value | OR (95% CI) | *P* value |
| Corticosteroid for based disease 1-40day [a] | 1.36 (1.15–1.60) | <0.001 | 1.39 (1.16–1.66) | <0.001 |
| Onset to First visit | 0.96 (0.89–1.05) | 0.405 | 0.97 (0.90–1.06) | 0.539 |
| Lactate dehydrogenase [b] | 1.44 (1.06–1.97) | 0.021 | 1.37 (0.90–1.88) | 0.049 |
| Lymphocyte count [c] | 0.89 (0.80–0.98) | 0.024 | 0.88 (0.78–0.98) | 0.023 |
| Hypoxia | 2.07 (0.64–6.73) | 0.228 | 1.76 (0.54–5.67) | 0.347 |
| Corticosteroid for PcP therapy 1-5day [d] | 1.03 (0.96–1.11) | 0.429 | 1.03 (0.95–1.12) | 0.442 |
| TMP-SMX for PcP therapy 1-5day [d] | 0.93 (0.84–1.02) | 0.108 | 0.93 (0.85–1.02) | 0.143 |

CI: confidence interval; OR: odds ration;

[a] per 0.1 mg/kg/day increment

[b] per 100 U/L increment

[c] per 100/mm$^3$ increment

[d] per 1mg/kg/day increment

In this study, we found that the prognostic impact of antimicrobials and steroids on treatment of non-HIV-PcP was less than that of corticosteroids for underlying disease. Three patients did not receive any antimicrobial therapy, but two of them died (the remaining patient resolved spontaneously; this was a patient with no underlying immunosuppressive disease). It is not unusual for eligible patients to have occult infection [48], but there are several reports of cases where eligible persons developed PcP [48–51], which generally has a poor prognosis. However, there are also reports of patients recovering without complications [50]. On the other hand, we report four cases of death without a history of steroid use. All cases had underlying hematologic disease (malignant lymphoma, aplastic anemia, myelofibrosis, and myelodysplastic syndrome). A longer the period from onset to treatment initiation was proportionally associated with a better prognosis. This may because many of the patients who started treatment early had rapid deterioration and respiratory failure, resulting in a poor prognosis. This suggests that immunity before disease onset has a greater impact on prognosis, suggesting that prevention is of paramount importance.

No treatment guidelines have been released for non-HIV-PcP, but this may be due to the diversity of patients' immune status. It is known that alveolar macrophages and CD4-positive T cells, play a major role in immunity against PcP [52]. Corticosteroids interfere with many pathways within the immune system and reduce the number and function of peripheral blood CD4-positive lymphocytes, clearly associated with a significant risk of PcP in non-HIV patients [53]. Corticosteroids have further been shown to impair alveolar macrophage function [54]. In a dexamethasone-induced immunosuppression model in rats, macrophages were found to be deficient in nitric oxide and hydrogen peroxide production even in Pc infection [55, 56], suggesting that the bactericidal capacity of Pneumocystis is problematic. These results may explain the poor prognosis of PcP associated with long-term steroid accumulation.

The current study has several limitations. Because of its retrospective nature, the diagnosis and treatment of PcP was not standardized; diagnosis was made using either only staining, PCR results or classical clinical criteria. Distinguishing PcP infection from colonization is challenging, and our exclusion criteria. Because PCR for PcP is not covered by healthcare insurance in Japan, many cases of suspected PcP were diagnosed by imaging and clinical findings of PcP based on cutoffs for $(1{\rightarrow}3)$ β-D-glucan elevation [57, 58], and many cases were likely missed. We excluded patients with clinical symptoms of PcP but negative PCR results. The conventional nested PCR targeting the mitochondrial large subunit rRNA of Pneumocystis [33] is one of the most common methods, but it may reduce sensitivity in cases of false negatives or low fungal load [59], leading to the exclusion of many cases. These limitations could be overcome by introducing a more sensitive and convenient assay [59]. We also had to exclude cases where estimated steroid use in the past 90 days could not be assessed and cases with no history of weight measurement, which may have affected our results due to selection bias. Also, only a rough assessment of respiratory impairment was possible. Additionally, too low a number of cases resulted in a situation where logistic regression analysis resulted in an overabundance of explanatory variables and insufficient test power. In the present study, it was impossible to determine whether continuous or intermittent dosing was riskier because it was not possible to distinguish whether daily dosing was pulsatile or continuous. Finally, Alb test data, which had been reported to contribute to PcP prognosis in previous studies, could not be evaluated due to the large number of missing values. Prospective studies and a larger number of cases must be considered in the future.

## Conclusions

In this propensity-matched cohort study, we found that corticosteroid administration in the 40-day period before the onset of PcP increased the mortality in a dose-dependent manner (per 0.1mg/kg/day increment, odds ratio 1.36 [95% CI = 1.16–1.66], $P<0.001$). Thus, long-term corticosteroid loading for treating the underlying disease strongly affects the prognosis of PcP, emphasizing the importance of appropriate prophylaxis especially in this population.

## Supporting information

**S1 File. Study database.**
(XLSX)

**S2 File. Basel corticosteroid dose database.**
(XLSX)

## Acknowledgments

We thank to Okayama Respiratory Disease Study Group for providing the case data. The lead author for this group is Nobuaki Miyahara (email address: miyahara@md.okayama-u.ac.jp.), and Members who contributed to our study are listed

[b]Department of Hematology, Oncology and Respiratory Medicine, Okayama University Graduate School of Medicine, Dentistry, and Pharmaceutical Sciences, Okayama, Japan.

Masahiro Tabata, Katsuyuki Hotta, Kadoaki Ohhashi, Kanmei Rai, Eiki Ichihara, Masanori Fujii, Go Makimoto, Kiichiro Ninbomiya

[d]Department of Respiratory Medicine, National Hospital Organization Minami-Okayama Medical Center, Okayama, Japan

Yasushi Tanimoto, Goro Kimura, Makoto Fujii

[e]Department of General Internal Medicine 4, Kawasaki Medical School, Okayama, Japan

Hiromichi Yamane, Nobuaki Ochi

[f]Department of Respiratory Medicine, National Hospital Organization Clinical Center, Iwakuni, Japan

Shoichi Kuyama, Kazuya Nishii, Takahiro Umeno

[g]Department of Respiratory Medicine, National Hospital Organization Okayama Medical Center, Okayama, Japan

Takuo Shibayama, Keiichi Fujiwara, Ken Sato, Akiko Sato, Hiromi Watanabe

## Author Contributions

**Conceptualization:** Kohei Miyake.

**Data curation:** Kohei Miyake, Satoru Senoo, Ritsuya Shiiba, Junko Itano, Goro Kimura, Tatsuyuki Kawahara, Tomoki Tamura, Kenichiro Kudo.

**Formal analysis:** Kohei Miyake.

**Investigation:** Kohei Miyake, Satoru Senoo, Ritsuya Shiiba, Junko Itano, Goro Kimura, Tatsuyuki Kawahara, Tomoki Tamura, Kenichiro Kudo.

**Methodology:** Kohei Miyake.

**Supervision:** Tetsuji Kawamura, Yasuharu Nakahara, Hisao Higo, Daisuke Himeji, Nagio Takigawa, Nobuaki Miyahara.

**Visualization:** Kohei Miyake.

**Writing – original draft:** Kohei Miyake.

**Writing – review & editing:** Kohei Miyake, Satoru Senoo, Ritsuya Shiiba, Junko Itano, Goro Kimura, Tatsuyuki Kawahara, Tomoki Tamura, Kenichiro Kudo, Tetsuji Kawamura, Yasuharu Nakahara, Hisao Higo, Daisuke Himeji, Nagio Takigawa, Nobuaki Miyahara.

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
