## [Decision Letter · Decision Letter 0]

3 Nov 2023

PONE-D-23-30570Pneumocystis jirovecii pneumonia mortality risk associated with preceding long-term steroid use for the underlying disease: a multicenter, retrospective cohort study.PLOS ONE

Dear Dr. Miyake,

Thank you for submitting your manuscript to PLOS ONE. After careful consideration, we feel that it has merit but does not fully meet PLOS ONE’s publication criteria as it currently stands. Therefore, we invite you to submit a revised version of the manuscript that addresses the points raised during the review process.

We look forward to receiving your revised manuscript.

Kind regards,

Benjamin M. Liu, MBBS, PhD, D(ABMM), MB(ASCP)

Academic Editor

PLOS ONE

[K.M. have received personal fees from Nippon Shinyaku Co td and AstraZeneca. N.T. have received grants and personal fees from Eli Lilly Japan, AstraZeneca, Daiichi-Sankyo Pharmaceutical, Chugai Pharmaceutical, Taiho Pharmaceutical, Pfizer Inc. Japan, Boehringer-Ingelheim Japan, and Ono Pharmaceutical; grants from Kyowa Hakko Kirin, Nippon Kayaku Co. Ltd., and Takeda Pharmaceutical Co. Ltd.; and personal fees from MSD and Bristol-Myers Squibb Company Japan outside the submitted work. None declared (S.S., R. S., J. I., G. K., T. K., T. T., K. K., T. K., Y. N., H. H., D. H., N. M.)]. 

4. One of the noted authors is a group [Okayama Respiratory Disease Study Group (ORDSG)]. In addition to naming the author group, please list the individual authors and affiliations within this group in the acknowledgments section of your manuscript. Please also indicate clearly a lead author for this group along with a contact email address.

Reviewers' comments:

Reviewer's Responses to Questions

**Comments to the Author**

1. Is the manuscript technically sound, and do the data support the conclusions?

Reviewer #1: Yes

2. Has the statistical analysis been performed appropriately and rigorously? 

Reviewer #1: Yes

3. Have the authors made all data underlying the findings in their manuscript fully available?

Reviewer #1: Yes

4. Is the manuscript presented in an intelligible fashion and written in standard English?

Reviewer #1: Yes

5. Review Comments to the Author

Reviewer #1: line 120, there are no reliable methods to differentiate PCP infection from colonization. Given high sensitivity of PCR, it is possible that there are pneumocysits infected patients without pulmonary symptoms but positive for PCR alone. More important, this exclusion made the authors miss some important cases. The authors are encouraged to re-word this portion.

Fig. 2, X axis should be "1-Specificity"

Fig.3, there are no signs of statistical analysis.

Line 331-334, the authors are encouraged to extend to discuss the dilemma and challenges of PCP diagnosis based on the following references.

Liu B, Totten M, Nematollahi S, Datta K, Memon W, Marimuthu S, Wolf LA, Carroll KC, Zhang SX. Development and Evaluation of a Fully Automated Molecular Assay Targeting the Mitochondrial Small Subunit rRNA Gene for the Detection of Pneumocystis jirovecii in Bronchoalveolar Lavage Fluid Specimens. J Mol Diagn. 2020 Dec;22(12):1482-1493. doi: 10.1016/j.jmoldx.2020.10.003. Epub 2020 Oct 15. Erratum in: J Mol Diagn. 2021 Apr;23(4):506. PMID: 33069878.

6. PLOS authors have the option to publish the peer review history of their article (what does this mean?). If published, this will include your full peer review and any attached files.

Reviewer #1: No

---

## [Author Response · Author response to Decision Letter 0]

18 Nov 2023

Response to Reviewers

Reviewer #1: line 120, there are no reliable methods to differentiate PCP infection from colonization. Given high sensitivity of PCR, it is possible that there are pneumocysits infected patients without pulmonary symptoms but positive for PCR alone. More important, this exclusion made the authors miss some important cases. The authors are encouraged to re-word this portion.

Response: We agree with the reviewer. We have removed the colonization detection part from the relevant section and added a discussion part about the PCP diagnostic dilemma.

Fig. 2, X axis should be "1-Specificity"

Response: I have corrected the relevant part. 

Fig.3, there are no signs of statistical analysis.

Response: We have added a description of the statistical analysis for the relevant part in the section.

Line 331-334, the authors are encouraged to extend to discuss the dilemma and challenges of PCP diagnosis based on the following references.

Liu B, Totten M, Nematollahi S, Datta K, Memon W, Marimuthu S, Wolf LA, Carroll KC, Zhang SX. Development and Evaluation of a Fully Automated Molecular Assay Targeting the Mitochondrial Small Subunit rRNA Gene for the Detection of Pneumocystis jirovecii in Bronchoalveolar Lavage Fluid Specimens. J Mol Diagn. 2020 Dec;22(12):1482-1493. doi: 10.1016/j.jmoldx.2020.10.003. Epub 2020 Oct 15. Erratum in: J Mol Diagn. 2021 Apr;23(4):506. PMID: 33069878.

Recponse: We appreciate your comment and the reference you suggested. We have revised the limitation section to acknowledge that many cases may be overlooked due to the sensitivity issue of PCR.

changes to the reference list

We have made changes because the publication issue has been decided.

(15. Miyake K, Kawamura T, Nakahara Y, Sasaki S. A single-center, person-month-based analysis of the risk of developing Pneumocystis pneumonia (PCP) in immunosuppressed non-HIV patients: Preventive effects of trimethoprim-sulfamethoxazole. ) J Infect Chemother. 2023;S1341-321X(23)00174-5. doi: 10.1016/j.jiac.2023.07.012. to J Infect Chemother. 2023; 29: 1097-1102.

We have added references.

59. Liu B, Totten M, Nematollahi S, Datta K, Memon W, Marimuthu S, Wolf LA, et al. Development and Evaluation of a Fully Automated Molecular Assay Targeting the Mitochondrial Small Subunit rRNA Gene for the Detection of Pneumocystis jirovecii in Bronchoalveolar Lavage Fluid Specimens. J Mol Diagn. 2020;22:1482-1493.

---

## [Editor Report · Decision Letter 1]

11 Jan 2024

Pneumocystis jirovecii pneumonia mortality risk associated with preceding long-term steroid use for the underlying disease: a multicenter, retrospective cohort study.

PONE-D-23-30570R1

Dear Dr. Miyake,

We’re pleased to inform you that your manuscript has been judged scientifically suitable for publication and will be formally accepted for publication once it meets all outstanding technical requirements.

Kind regards,

Benjamin M. Liu, MBBS, PhD, D(ABMM), MB(ASCP)

Academic Editor

PLOS ONE
---

## [Editor Report · Acceptance letter]

31 Jan 2024

PONE-D-23-30570R1 

PLOS ONE

Dear Dr. Miyake, 

I'm pleased to inform you that your manuscript has been deemed suitable for publication in PLOS ONE. Congratulations! Your manuscript is now being handed over to our production team.

Kind regards, 

on behalf of

Dr. Benjamin M. Liu 

Academic Editor

PLOS ONE